# Improved Central Nervous System Symptoms in People with HIV without Objective Neuropsychiatric Complaints Switching from Efavirenz to Rilpivirine Containing cART

**DOI:** 10.3390/brainsci9080195

**Published:** 2019-08-09

**Authors:** Jaime H. Vera, Margherita Bracchi, Jasmini Alagaratnam, Julianne Lwanga, Julie Fox, Alan Winston, Marta Boffito, Mark Nelson

**Affiliations:** 1Department of Global Health and Infection, Brighton and Sussex Medical School and Brighton and Sussex University Hospitals NHS Trust, London BN1 9PX, UK; 2Chelsea and Westminster NHS Foundation Trust, London SW10 9NH, UK; 3Imperial College London, St Mary’s Campus, London W2 1PG, UK; 4Guy’s and St Thomas’ NHS Trust, London SE1 7EH, UK

**Keywords:** HIV, efavirenz, rilpivirine, Central nervous system

## Abstract

**Objective:** Occult central nervous system (CNS) symptoms not recognized by people living with HIV (PLWH) receiving efavirenz or their clinicians could occur and impact people’s quality of life. The aim of this study was to determine whether CNS parameters improve in PLWH when switching from efavirenz to rilpivirine. **Methods:** PLWH receiving tenofovir disoproxil fumarate, emtricitabine, efavirenz (Atripla™) with undetectable HIV RNA, and no CNS symptoms were switched cART to tenofovir disoproxil fumarate, emtricitabine, rilpivirine (Eviplera™). CNS parameters including sleep, anxiety, and depressive symptoms were evaluated using patient-reported outcome measures at baseline, 4, 12, and 24 weeks after switching therapy. A median CNS score was derived from the sum of CNS toxicities of all the grades collected in the study questionnaires. Cognitive function was assessed using a computerized test battery. **Results:** Of 41 participants, median age was 47 years, Interquartile range (IQR) 31, 92% were male and 80% were of white ethnicity. A significant reduction in total CNS score (10 to 7) was observed at 4 weeks (*p* = 0.028), but not thereafter. Significant improvements in sleep and anxiety were observed 4, 12 and 24 weeks after switching therapy (*p* < 0.05). No significant change in global cognitive scores was observed. **Conclusions:** Switching from efavirenz to rilpivirine based regimens in virologically suppressed PLWH without perceived CNS symptoms was well tolerated and slightly improved overall CNS symptoms.

## 1. Introduction

Antiretroviral toxicity is a common cause for therapy modification with around a quarter of individuals on first-line combination antiretroviral therapy (cART) changing treatment due to the onset of toxicity including central nervous system (CNS) toxicity [1,2]. Efavirenz is a highly efficacious and convenient (once daily dosing) non-nucleoside reverse transcriptase inhibitor (NNRTI) that is still being used as a first-line therapy in combination with other antiretrovirals as a single tablet regimen atripla (tenofovir disoproxil fumarate, emtricitabine, efavirenz) or at doses of 600 mg and 400 mg in low income settings [3,4,5]. CNS toxicity is a common side effect of efavirenz and although most CNS toxicity with efavirenz resolves in 2–4 weeks [6], between 20% to 40% of patients still experience ongoing neuropsychiatric symptoms such as anxiety, depression and abnormal dreams after several weeks or even years after the initiation of efavirenz therapy [7,8,9,10]. Furthermore, switching from efavirenz-containing regimes to efavirenz free cART has been associated with improved CNS toxicity in several studies [8,11,12,13]. More recently, Efavirenz use has been associated with poorer cognitive function in several cohort studies [14,15]. Hakkers et al. recently reported a randomized controlled trial where an improvement in cognitive function after switching from efavirenz to rilpivirine based cART was observed in asymptomatic patients on effective cART [16]. In contrast, another randomized controlled trial of people with HIV on efavirenz switching to rilpivirine did not show a significant improvement in cognitive function or patient reported outcomes of depression and anxiety [17]. Similarly, Payne et al. showed that switching people with HIV on efavirenz for at least 6 months to a protease inhibitor combination was not associated with changes in cognitive function or brain imaging parameters [18].

Patients on efavirenz based cART might not actively report CNS symptoms attributable to efavirenz. However, concern exits regarding occult CNS toxicities which could have a deleterious effect on quality of life and cognitive function after years of exposure to efavirenz. Rilpivirine is a second-generation NNRTI which has demonstrated to be non-inferior to efavirenz in treatment naïve HIV individuals with a viral load ≤100,000 HIV-1 RNA copies/mL [19,20,21]. Rates of neuropsychiatric side effects and discontinuations are significantly lower with rilpivirine when compared to efavirenz (16% versus 37%) (1% versus 5%), respectively [19,20]. Given the recognized effects of efavirenz on neuropsychiatric symptoms, we hypothesize that switching people with stable HIV on efavirenz without objective CNS toxicity to rilpivirine containing cART leads to improvement in CNS symptoms, health-related quality of life (QoL), sleep, mood and cognitive function. 

## 2. Methods

### 2.1. Participants and Data Collection

HIV-1 positive adults who had received at least 12 weeks of tenofovir disoproxil fumarate, emtricitabine, efavirenz without documented history in medical records of neuropsychiatric or CNS symptoms or any significant self-perceived objective CNS symptoms at the time of recruitment were enrolled in a phase IV, open label single arm study in 4 UK sites (Chelsea and Westminster Hospital, St Mary’s Hospital, Guy’s and St Thomas’s Hospital in London and the Royal County Hospital in Brighton) between January 2016 and January 2017. Patients were required to have a plasma HIV RNA <40 copies/mL (the target to define a patient as undetectable in the UK) and CD4 cell count greater than 50 cells/μL at screening. Subjects were excluded if they had previous exposure to rilpivirine, acute or chronic viral hepatitis, evidence of HIV-drug resistance mutations prior to commencing cART or due to use of disallowed concomitant medication (as per the summary of product characteristics for tenofovir disoproxil fumarate, emtricitabine, rilpivirine). Subjects were also ineligible if they had current self-reported recreational drug use or excessive alcohol consumption. All subjects gave written informed consent. The study was approved by the research ethics committee, the UK regulatory authority and medicines and health care products regulatory agency (MRHA) and registered with the EudraCT trials database (2014-002284-15).

On day 1, eligible subjects were switched from tenofovir disoproxil fumarate, emtricitabine, efavirenz (Atripla) to tenofovir disoproxil fumarate, emtricitabine, rilpivirine (Eviplera) for the duration of the study period (24 weeks). Patients were followed up at 4, 12 and 24 weeks.

### 2.2. Study Assessments

At each study visit, the patients underwent blood sampling for plasma HIV-RNA, CD4, CD8 cell count, biochemistry (hepatic, renal and bone profiles), haematology (full blood count and differential), fasting lipids ([total, high-density lipoprotein (HDL) and low-density lipoprotein (LDL) cholesterol, total: HDL cholesterol ratio and triglycerides], fasting glucose and urine macro-analysis.

### 2.3. Neuropsychiatric Symptoms and Patient Reported Outcome Measures

Central nervous symptoms were assessed using a CNS side effects questionnaire based on the summary of products characteristics (SPC) and the division of AIDS grading scale (DAIDS) for grading the severity of adverse events [22]. Study participants were specifically questioned by a research nurse or doctor at each study visit about any objective neuropsychiatric and CNS adverse events and the medical notes were reviewed to ensure that any CNS symptoms had not been documented by other health care professionals. CNS symptoms are listed in Table 1. Each symptom was graded on a 4-point scale (0 = none, 2 = mild, 3 = moderate, 4 = severe). CNS adverse events were described as the proportion of patients with any grade 2–4 adverse events and individual grade 2–4 adverse events; the median number of grade 2–4 CNS adverse events were also calculated. A “CNS score” was calculated based on the sum total of all grades of the CNS adverse events and these were transformed so that the scores ranged from 0 to 100. A maximum possible score is 100 (severe) and a minimum possible score is 0 (none). 

Sleep, anxiety and depression symptoms were measured using the Pittsburgh Sleep Questionnaire [23] and the Hospital and Anxiety and Depression questionnaires, respectively [24]. 

Quality of life was measured at 4, 12 and 24 weeks using the EuroQoL (EQ-5D-5L)—a standardized instrument to measure health-related quality of life. A EQ VAS (visual analogue scale for health related quality of life) score was calculated, where a maximum score of 100 corresponds to the best-reported health and a minimum score of 0 represents the worst reported health [25].

### 2.4. Cognitive Testing

Cognitive testing was performed at 4 and 24 weeks using CogState (CogState Ltd., Melbourne, Australia), a computerized battery previously validated in people living with HIV (PLWH) [26]. The battery evaluates several cognitive domains including speed, visual attention, accuracy, visual learning, working memory, verbal learning, associate learning and executive function.

### 2.5. Statistical Analysis

The primary endpoint of the study was change in neuropsychiatric and central nervous system (CNS) parameters in patients without perceived CNS symptoms after 4 weeks of switching from efavirenz to rilpivirine based cART.

All participants who were enrolled and subsequently baselined into the study formed part of the statistical analysis. All summary statistics of the quantitative data are presented with point estimates and indication of variability in data where hyper-geometric distributed data are presented as a median with an inter-quartile range and Gaussian normal data are presented as a mean with a standard deviation. Qualitative data are presented as numbers and percentages. Within study time point changes from baseline for non-parametric data such as CNS scores and patient reported outcome scores were tested using the Wilcoxon signed rank test. 

CogState results from PLWH within the study participants were compared with standardized normative data of the general population provided by Cogstate. Raw scores from each of the cognitive domains tested were transformed to *z*-scores using the CogState manufacturers’ normative data adjusted for age by subtracting the mean and dividing by the standard deviation (SD) of the test scores in the reference population. Summary *z*-scores were corrected for multiple comparisons and then calculated by averaging *z*-scores of all the tests where normative data were available. Characteristics of CogState battery tested over time are described as composite age-adjusted *z*-score weighted mean. Due to the multiplicity of data per patient, within changes over time in the age-adjusted composite scores between baseline and week 4 and week 24 were tested using a weighted paired *t*-test. All statistical analyses were performed using SAS version 9.4 statistical software (SAS Institute, Cary, NC, USA) and all *p*-values presented are two-tailed.

## 3. Results

Forty-one patients who were predominantly male (92%) and of white ethnicity (85%) were enrolled. Four participants withdrew from the study. Two participants left at week 4 and 12 because they were unable to complete the 24 weeks due to changes in personal circumstances, and two other participants withdrew between 12 and 24 weeks because they were moving away from the UK indefinitely. A total of 37 individuals completed all the study procedures. Median age (range) was 47 years [27,28,29,30,31,32,33,34] and median duration on Atripla at baseline was 5 years [2,3,4,5,6,7]. Median baseline CD4 count was 563 cells/μL (IQR 465-679) and all patients including those that completed the study had a HIV RNA <40 copies/mL throughout the study period (Table 2).

### 3.1. Change in Central Nervous Symptoms and Patient Reported Outcomes for Sleep, Anxiety and Depression

Rates of CNS symptoms at baseline, week 4, 12 and 24 are presented in Table 1. All patients had at least one grade 2–4 CNS symptom at baseline, with insomnia or sleepiness and abnormal dreams as the most common symptoms. The Median Interquartile range (IQR) CNS score at baseline was 10 (2–23). At week 4, the median score was 7 (3–13), representing a significant decline in any grade 2–4 CNS symptoms compared with baseline: median change 0 (0–7); *p* = 0.028 (primary endpoint). In terms of individual grade 2–3 CNS symptoms, at week 4, there was a 20% decline in the proportion of patients experiencing dizziness and insomnia, a 22% decline in abnormal dreams and a 10% reduction in headaches compared to baseline (Table 1). In contrast, there was a 20% increase in somnolence at 4 weeks after starting rilpivirine. There were no significant changes in total and individual CNS scores from baseline apart from improvement in abnormal dreams, which continued at week 12 and 24 following the switch to rilpivirine based cART (Table 1). Regarding patient reported outcomes, at week 4, there was a 2% decline in anxiety symptoms (range: 0–7; *p* = 0.001) and a 4.8% decline in sleep symptoms (range: 0–9.5; *p* = 0.001) (Figure 1). Improvements in sleeping symptoms and anxiety were also observed at week 12 and 24. No significant changes to depression scores from baseline were observed at weeks 4, 12 or 24 (Figure 1). 

### 3.2. Change in Health-Related Quality of Life

A significant improvement in health-related quality of life was observed with EQ-VAS scores increasing by 2% at week 4 (mean change: 2 (−10 to 0); *p* = 0.003) following the switch to rilpivirine. This improvement was still present at week 12 and 24 (Figure 1).

### 3.3. Change in Cognitive Function

Cognitive function was within the normal range for all patients at baseline when compared to the normalised data provided (HIV negative individuals aged 20 to 70 years old) by the manufacturer (CogState). No significant changes in global cognitive function were observed at 4 or 24 weeks after starting TDF/FTC/RPV. However, when looking at individual cognitive domains, significant improvements in executive function, attention and learning were observed at week 24 after switching to rilpivirine (Table 3).

### 3.4. Virological and Immunological Efficacy 

All participants in this study maintained viral suppression (less than 50 copies/mL) at all visits. No significant changes in CD4 and CD8 T cell counts were observed at weeks 12 and 24 (*p* > 0.1).

### 3.5. Lipids

Although 39% [16] of participants were on medications to reduce lipids, there was a significant reduction in total cholesterol, LDL-cholesterol, triglycerides and HDL-cholesterol at weeks 4, 12 and 24 (*p* < 0.05 for all measurements). A significant reduction in the total cholesterol/HDL ratio was only observed at week 12 (*p* = 0.005)

### 3.6. Non-Central Nervous System Adverse Events and Laboratory Abnormalities

TDF/FTC/RPV was well tolerated. There were no new grade 2–4 adverse events and no significant changes in liver, renal and other laboratory parameters at weeks 4, 12 and 24.

## 4. Discussion

This study suggests that switching efavirenz to rilpivirine containing cART in virologically suppressed people with HIV without perceived objective CNS toxicity slightly improves neuropsychiatric symptoms, cognitive function and health-related quality of life, but does not improve overall CNS cognitive function. Switching to rilpivirine containing cART improved overall CNS symptoms at 4 weeks (primary outcome), though improvement in the total CNS score was not observed at 12 and 24 weeks. When looking at individual CNS symptoms, the greatest improvement was observed in sleep associated symptoms and anxiety. We found that sleep quality and nervousness that were not perceived as side effects of efavirenz were significantly better up to 24 weeks after switching to rilpivirine. Cognitive function was found to be preserved in all participants at baseline. While switching to rilpivirine did not result in significant changes to global cognitive function, there were improvements in executive function, attention and learning after 24 weeks. 

CNS symptoms have consistently been reported in 40% to 60% of patients who initiate efavirenz and it is a common reason for discontinuation. However, the majority of CNS side effects appear early and are usually resolved within the first 6 weeks of treatment. However, there is a proportion of patients where neuropsychiatric symptoms, such as sleep difficulties, anxiety and depression, persist for much longer periods, even years after initiation of efavirenz therapy [9,35,36]. The reasons for this phenomenon could be related to interindividual variations in the way the drug is metabolized in the presence or absence of a polymorphism in CYP2BE. Individuals carrying alleles G/T or T/T of CYP2BE are less efficient at metabolizing the drug than doses with the allele G/G, and therefore are more likely to have a high plasma concentration or longer plasma high life of efavirenz [37]. Higher plasma concentrations of efavirenz have been associated with greater CNS symptoms [27,38]. More recently, a randomized, double blind study reported that a reduced dose of 400 mg of efavirenz was non-inferior to the standard dose of 600 mg and was associated with a lower frequency of CNS adverse events [28]. In our study, we were unable to test for individual differences in CYP2BE genotype, and all participants were on the standard 600 mg dose of efavirenz, which could explain why they were more likely to benefit from switching. There are also the inherent issues with bias associated with patient reported outcomes measurements in open trials as patient’s knowledge of the treatment received might influence their view and reporting of their symptoms. 

The effect of efavirenz on cognitive function remains controversial, with some observational studies showing an increased rate of cognitive impairment associated with efavirenz use [14] and others showing no changes in cognitive function after efavirenz withdrawal [17,18]. Neurotoxicity data from animal and in vitro studies suggests that efavirenz might have a detrimental effect on human brain cells [29,30,31] and therefore, another reason why some patients might not be showing clinical impairment is because of individual differences in brain reserve. 

We observed an increase in depression symptoms using the CNS questionnaire at week 24. In contrast, no significant changes in depression scores were found when using the hospital and anxiety depression scale at week 24. The discrepancy of the results might be due to differences in the sensitivity of the tools employed to detect depression symptoms. As rilpivirine has been associated with depression symptoms [32], further research into the prevalence of depressive symptoms in patients taking rilpivirine based combinations is recommended.

In this study, we also described an improvement in lipids with significant reductions in total and LDL-cholesterol after 24 weeks of rilpivirine, which is in keeping with previous studies showing a significant reduction in lipids following a switch from efavirenz to rilpivirine based cART [33]. Efavirenz is known to be associated with increases in lipid parameters [34] and rilpivirine, less so. These results confirm that rilpivirine may be an option to switch to for metabolic considerations.

There are several limitations to this study. First, it is possible that the patients recruited for the study had significant self-perceived CNS symptoms that were not considered to be attributed to efavirenz during screening and therefore enhancing the beneficial effect of switching to rilpivirine based cART. Most patients that were recruited for this study had been on efavirenz based cART for many years without any documented CNS complaints in their medical notes before enrolment. To further minimise bias during recruitment, the research doctor or nurse had to assess if the complaints were objective or subjective and attributable to efavirenz or any other cause. Second, the improvement in some cognitive function domains could be explained by the learning effect of repeat testing in such a short interval. Finally, the lack of a control arm, a small sample size and the fact that most participants were male and from white ethnicity limits the generalisability of the results.

## 5. Conclusions

Several studies have shown that in patients with symptomatic CNS side effects attributed to efavirenz, switching therapy leads to improvements in CNS toxicity [8,34]. What our study has shown is that in patients without perceived CNS symptoms, switching away from efavirenz might yield improvements in patient reported outcomes of sleep, anxiety and health-related quality of life, which are important clinical outcomes for stable patients on cART.

## Figures and Tables

**Figure 1 brainsci-09-00195-f001:**
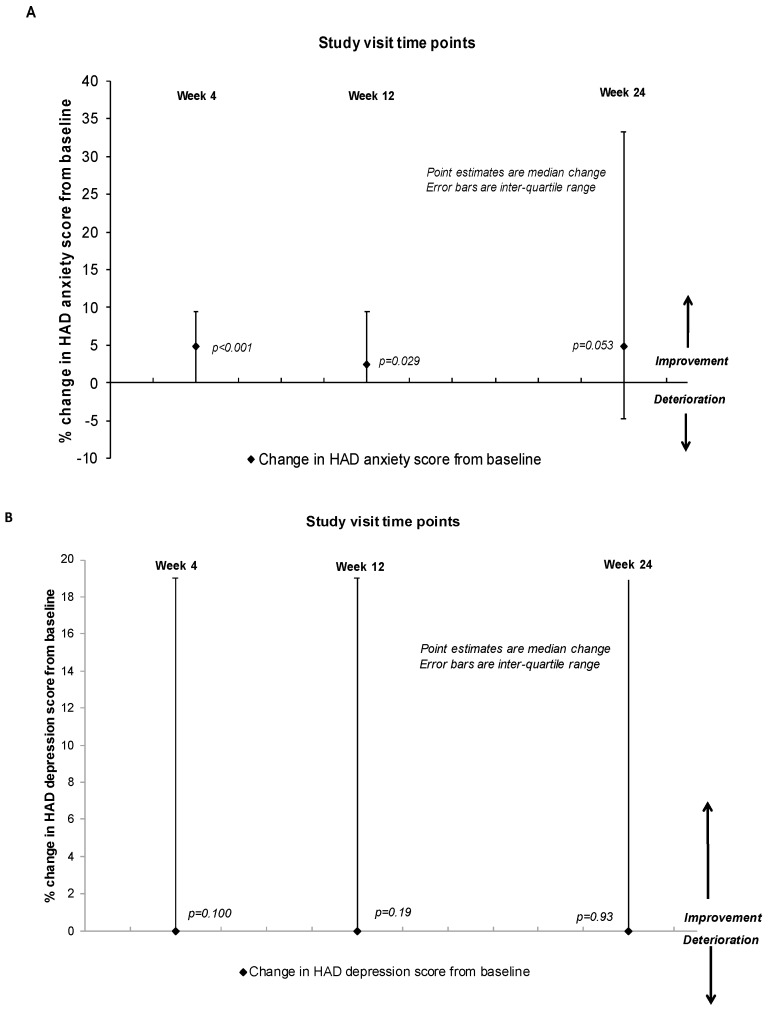
Change in patient reported outcomes for anxiety (**A**), depression (**B**), sleep (**C**) and health related quality of life scores (**D**) at 4, 12 and 24 weeks from baseline. HAD = hospital anxiety and depression scale, QoL = quality of life. Point estimates are median change; error bars are IQR change following the Wilcoxon rank test.

**Table 1 brainsci-09-00195-t001:** Rates of grade 2–4 central nervous system (CNS) side effects at week 4, 12 and 24 (*p* values correspond to the difference at each time point compared with event rates at baseline).

CNS Side Effects	Baseline *n* = 41	4 Weeks *n* = 40		12 Weeks *n* = 39		24 Weeks *n* = 37	
			Median change * *p* value		Median change * *p* value		Median change * *p* value
**Total CNS score, median (IQR)**	10 (2–23)	7 (3–13)	0.028	10 (3–17)	0.064	7 (3–17)	0.152
**Dizziness, *n* (%)**	10 (24)	2 (5)	0.005	5 (12)	0.096	5 (13)	0.103
**Depression/Low mood, *n* (%)**	14 (34)	14 (35)	0.65	16 (41)	0.25	17 (46)	0.04
**Insomnia/sleepiness, *n* (%)**	23 (56)	15 (37)	0.05	15 (38)	0.134	19 (51)	0.73
**Anxiety/nervousness, *n* (%)**	14 (34)	12 (30)	0.31	16 (41)	0.56	15 (40)	0.78
**Confusion, *n* (%)**	3 (7)	7 (17)	0.20	3 (7)	0.99	5 (13)	0.48
**Impaired concentration, *n* (%)**	14 (34)	15 (37)	0.52	11 (28)	0.52	11 (29)	0.48
**Somnolence, *n* (%)**	9 (22)	16 (40)	0.02	16 (41)	0.02	10 (27)	0.706
**Aggressive mood behaviour, *n* (%)**	9 (22)	7 (17)	0.65	5 (12)	0.08	7 (18)	0.56
**Abnormal dreams, *n* (%)**	22 (53)	12 (30)	0.003	10 (25)	0.002	8 (21)	0.001
**Headache, *n* (%)**	12 (29)	7 (17)	0.04	7 (18)	0.15	7 (18)	0.20

* Wilcoxon rank test.

**Table 2 brainsci-09-00195-t002:** Demographic characteristics at baseline.

Demographic/Clinical Parameter	Total
Number of subjects	41
Age, years, median, interquartile range (IQR)	47.3 (31 to 68)
Male gender, *n* (%)	38 (92)
Ethnicity, *n* (%)	
White.	36 (85)
Afro-Caribeean	4 (10)
Other	2 (5)
Years on Atripla, median (IQR)	5 (2–7)
Years since HIV diagnosis median (IQR)	11(8–18)
Baseline CD4 cell count (cells/μL), median (IQR)	563 (465–679)
Baseline plasma HIV RNA level < 50 copies/mL, *n* (%)	41 (100)

**Table 3 brainsci-09-00195-t003:** Change in cognitive function 4 and 24 weeks after switching to Eviplera. P values correspond to the median change in absolute scores for all cognitive domains from baseline to week 4 and 24, respectively.

Cognitive Tests	Cognitive Domain	Baseline *n* = 41	4 Weeks *n* = 40	24 Weeks *n* = 37
**OCL (one card learning)** **Higher score = better performance** **Median (IQR)**	Learning	0.95 (0.8 to 1)	0.98 (0.8 to 1)	1.1 (0.9 to 1)
			0.02 (−0.08 to 0.09) *p* = 0.547	0.05 (−0.0 to 0.14) *p* = 0.035
**ONB (one back memory)** **Higher score = better performance** **Median (IQR)**	Working memory	1.32 (1.1 to 1.39)	1.32 (1.1 to 1.39)	1.39 (1.32 to 1.39)
			−0.05 (−0.17 to 0.16) *p* = 0.55	0.07 (0.00 to 0.20) *p* = 0.164
**TWOB (two back memory)** **Higher score = better performance**	Working memory	1.19 (1.11 to 1.33)	1.27 (1.02 to 1.33)	1.33 (1.19 to 1.40)
**Median (IQR)**			0.05 (−0.05 to 0.17) *p* = 0.55	0.07 (0.00 to 0.17) *p* = 0.164
**DET (Detection)** **Lower score = better performance**	Speed	2.51 (2.47 to 2.63)	2.57 (2.47 to 2.65)	2.57 (2.47 to 2.61)
**Median (IQR)**			−0.01 (−0.03 to 0.06) *p* = 0.581	0.01 (−0.02 to 0.07) *p* = 0.723
**IDN (Identification)** **Lower score = better performance**	Attention	2.71 (2.67 to 2.76)	2.71 (2.69 to 2.82)	2.75 (2.68 to 2.81)
**Median (IQR)**			−0.01 (−0.03 to 0.05) *p* = 0.24	0.02 (−0.02 to 0.08) *p* = 0.023
**GML (Groton maize learning tests)** **Lower score = better performance** **Median (IQR)**	Executive function	13 (7 to 24)	10 (6 to 20)	10 (6 to 17)
			−7 (−19 to 5) *p* = 0.147	−8 (−25 to 7) *p* = 0.011
**SETSID (Intradimensional shift)** **Lower score = better performance** **Median (IQR)**	Executive function	1.05 (1.00 to 1.29)	1.08 (1.00 to 1.29)	1.19 (1.02 to 1.29)
			−0.03 (−0.09 to 0.19) *p* = 0.347	0.10 (−0.05 to 0.23) *p* = 0.004
**Global cognitive Z score** **Z score based on normative data** **Median (IQR)**	All domains	−0.12 (−0.78 to 0.47)	−0.06 (−0.78 to 0.72) *p* = 0.758	0.04 (−0.37 to 0.58) *p* = 0.497

Improved central nervous system symptoms in people with HIV without overt neuropsychiatric complaints switching from efavirenz to rilpivirine containing cART.

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
