# Peer review of "Improved Central Nervous System Symptoms in People with HIV without Objective Neuropsychiatric Complaints Switching from Efavirenz to Rilpivirine Containing cART"

_brainsci, 2019, doi:10.3390/brainsci9080195_

Round 1

Reviewer 1 Report

General Comments

Bias- essentially male and white population, acknowledge results may not be applicable to wider HIV population.

Confounding- The absence of controls means it is extremely difficult to attribute improvements in CNS functioning to Riplrivirine alone. What is to say that a similar group of patients will not experience improvement in CNS functioning over time on EFV? Some further discussion of potential confounders and alternative explanation for the findings will be helpful.

The numbers presented in the results do not match the number presented in Table 2. I is suggested the authors go through the results and ensure numbers in the text match those in the table. It is also convention to present estimates to at least 1 decimal place.

Line 48: Missing "to" between switching and riplivirine

Line 60: stable HIV or HIV, stable

Line 89: Has the SPC as a basis for such  questionnaire been piloted in this or similar populations, ref if so. similarly reference use of DAIDS grading scale in previous research in similar population

Lines 148 and 149 on sleep, anxiety and QoL should be moved to section 3.2

Line 151: would appear results for anxiety and sleep have been switched round.

Line 153: Depression score is not represented in fig 1

Line 159: Describe what population the neurcognitive tests are normalised to

Table 3: is insufficiently labelled. It is not clear to me what all the numbers represent. I imagine some are z scores but does not say in the table label. In addition the heading says that data is presented for 4, 12 and 24 weeks, however the table only contains results for 4 and 24 weeks. 

Author Response

Reviewer 1

General Comments

Bias- essentially male and white population, acknowledge results may not be applicable to wider HIV population.

Response:

A sentence in the discussion has been added discussing bias due to gender and ethnicity and other limitations of the study. 

Confounding- The absence of controls means it is extremely difficult to attribute improvements in CNS functioning to Riplrivirine alone. What is to say that a similar group of patients will not experience improvement in CNS functioning over time on EFV? Some further discussion of potential confounders and alternative explanation for the findings will be helpful.

Response:

We have complemented the discussion with an alternative explanation for the findings and added supporting references

CNS symptoms have consistently been reported in 40% to 60 % of patients who initiate efavirenz and it is a common reason for discontinuation. However, the majority of CNS side effects appear early and are usually resolved within the first 6 weeks of treatment. There is a proportion of patients were neuropsychiatric symptoms such as sleep difficulties, anxiety and depression persist for much longer periods, even years after initiation of efavirenz therapy (9, 27, 28). The reasons for this phenomenon could be related to interindividual variations in the way the drug is metabolised in the presence or absence of a polymorphism in CYP2BE. Individuals carrying alleles G/T or T/T of CYP2BE are less efficient at metabolising the drug than dose with the allele G/G and therefore are more likely to have a high plasma concentration or longer plasma high life of efavirenz (29). Higher plasma concentration of efavirenz has been associated with CNS symptoms (30)(31). More recently a randomized, double blind study reported that a reduced dose of 400mg of efavirenz was non inferior to the standard dose of 600mg and was associated with a lower frequency of CNS adverse events (32). In our study we were unable to test for individual differences in CYP2BE genotype, and all participants were on the standard 600mg dose of efavirenz which could explain why there were more likely to benefit from a medication switch. There is also the inherent issues with bias associated with patient reported outcomes measurements in open trials as patient’s knowledge of the treatment received might influence their view and reporting of their symptoms.”.

The numbers presented in the results do not match the number presented in Table 2. I is suggested the authors go through the results and ensure numbers in the text match those in the table. It is also convention to present estimates to at least 1 decimal place.

Response:

Table 2 has been corrected and improved by adding information on statistical tests performed to calculate changes in CNS scores

Line 48: Missing "to" between switching and riplivirine

Response:

Amended

Line 60: stable HIV or HIV, stable

Response:

Amended

Line 89: Has the SPC as a basis for such questionnaire been piloted in this or similar populations, ref if so. similarly reference use of DAIDS grading scale in previous research in similar population

The SPC was used to inform the questionnaire by identifying the most common adverse reactions reported by patients and participants of clinical trials using the European SPC reporting system, as such we didn’t consider necessary to conduct a pilot study in people with HIV.

DAIDS grading scale has been validated in multiple clinical trials in HIV and it is regularly used to report adverse events associated with trials in HIV. We have added an additional reference that describes the tools framework and validity in the HIV population.

https://www.niaid.nih.gov/research/daids-clinical-research-event-reporting-safety-monitoring

Lines 148 and 149 on sleep, anxiety and QoL should be moved to section 3.2

Response:

The information has been moved and the section has been amended accordingly. Line 189 to 197

Line 151: would appear results for anxiety and sleep have been switched round.

Response:

Amended

Line 153: Depression score is not represented in fig 1

Response

It has not been presented because results were shown to be not significant. This information has now been added to the text and figure 1

Line 159: Describe what population the neurocognitive tests are normalised to

Response

Information on the normalise data has been added. “HIV negative individuals age 20 to 50 years old)

Table 3: is insufficiently labelled. It is not clear to me what all the numbers represent. I imagine some are z scores but does not say in the table label. In addition the heading says that data is presented for 4, 12 and 24 weeks, however the table only contains results for 4 and 24 weeks. 

Response:

The table has been modified and amended as recommended

Reviewer 2 Report

In this manuscript, J.H. Vera et al. analyzed the effect of switching from efavirenz to rilpirivirine in terms of occult CNS symptoms in virologically suppressed people with HIV (PWH). Patients considered neuropsychiatrically unhealthy were carefully excluded from the study. The authors observed potential changes in CNS side effects in the PWH. The given methods and data seem reasonable. The text is well written and easy to read. Despite the overall weakness of the data as appropriately discussed in the main text, I suppose the results would be worth publication as a clinical article. It is highly recommended to add individual time course data and correlation analyses (at weeks 4, 12 and 24) to improve logical connections between the tables and figure. My concerns are listed below.

1.      In Table 2, the results are intriguing despite the statistical weakness.

a)      The statistical test method for the total CNS scores could be described.

b)      The total CNS scores and some CNS symptoms (dizziness, insomnia, somnolence, headache) showed significant changes at week 4 but not at week 24, indicating the switching from efavirenz to rilpirivirine had limited impact on these symptoms. There were non-significant declines in the frequencies of Total CNS scores and the frequencies of dizziness and headache at week 24 compared to the baseline, which is interesting but not definitive.

c)      There was a significant increase in the frequency of depression at week 24 compared to the baseline. Could that indicate a potential side effect of rilpirivine? For reference, Freeman D, Levenson J. Rilpivirine and Depression. Psychosomatics. 2015 Nov-Dec;56(6):711-2. doi: 10.1016/j.psym.2015.06.001. Epub 2015 Jun 6. PubMed PMID: 26674487.

d)      It seems that cessation of efavirenz dramatically reduced the frequency of abnormal dreams. I suppose it might be helpful to clarify how this affected or correlated with the symptoms, scores, and shown in Table 2 and Figure 1 (insomnia, somnolence, total CNS, sleep, anxiety, QoL) as well as the cognitive functions described in Table 3.

2.      In Figure 1, the results are impressive. Could the authors better clarify the causes or correlates of the improved sleep/anxiety/QoL scores? How could the Table 2 results (statistically weak) be logically connected to the Fig 1 results (not weak)? Readers could be interested in individual sample data.

3.      Because the sample size is relatively small for a clinical trial results, I could recommend showing some data with plots. This could be highly effective for paired data.

4.      If statistical analysis was performed with paired tests as described in Methods, I suppose the sample size would be the same regardless of time points. However, the sample size seems to change at different time points (baseline, 4 weeks, 12 weeks, 24 weeks) in Tables 2 and 3. What happens if the unpaired samples are excluded from the statistics described in Tables 2 and 3?

5.      Figure/Table legends could give more information so that readers will be able to better understand these without referring to the main text.

6.      Discussion seems to be too short and limited. For example, the authors could better discuss the mechanisms for the improved lipid metabolism as described in page 4 lines 167-171, and its impact on the other observations described in Tables 2, 3, and Figure 1.

7.      I suppose “IQR” mean “interquartile range”, but the abbreviation is not defined in the text.

8.      In page 2 line 70, “be” seems unnecessary.

9.      In page 3 line 111, some words seem to be missing around “change”.

10.   Better use of commas could further improve the readability of the text.

11.   In Authors’ contributions, I could not identify MS, JJ, and JB. Is MB M. Bracchi or M. Boffito?

Author Response

Reviewer 2

The statistical test method for the total CNS scores could be described.

Response:

We have added further information in the statistical section on how within study time point changes from baseline for in non-parametric data such as CNS scores were tested using Wilcoxon signed rank test. Table 2 headings have also been modified to specify the type of statistical analysis performed.

The total CNS scores and some CNS symptoms (dizziness, insomnia, somnolence, headache) showed significant changes at week 4 but not at week 24, indicating the switching from efavirenz to rilpirivirine had limited impact on these symptoms. There were non-significant declines in the frequencies of Total CNS scores and the frequencies of dizziness and headache at week 24 compared to the baseline, which is interesting but not definitive.

Response.

We agree with the reviewer that no significant changes were observed at week 24. The primary outcome for this study was change in CNS scores at week 4. The rationale was to try reducing reporting bias of symptoms as in open trials patient knowledge of the treatment received could influence their view and reporting of their symptoms. We have added a paragraph in the discussion about this issue and other important limitations of the study

There was a significant increase in the frequency of depression at week 24 compared to the baseline. Could that indicate a potential side effect of rilpirivine? For reference, Freeman D, Levenson J. Rilpivirine and Depression. Psychosomatics. 2015 Nov-Dec;56(6):711-2. doi: 10.1016/j.psym.2015.06.001. Epub 2015 Jun 6. PubMed PMID: 26674487.

Response

We have added a paragraph describing the result including the suggested reference.We observed an increase in depression symptoms using the CNS questionnaire at week 24. In contrast, no significant changes in depression scores were found when using the hospital and anxiety depression scale at week 24. The discrepancy of the results might be due to differences in the sensitivity of the tools employed to detect depression symptoms. As rilpivirine based cART have been associated with depression symptoms (36)further research into the prevalence of depressive symptoms in patients taking rilpivirine based cART is recommended”.

It seems that cessation of efavirenz dramatically reduced the frequency of abnormal dreams. I suppose it might be helpful to clarify how this affected or correlated with the symptoms, scores, and shown in Table 2 and Figure 1 (insomnia, somnolence, total CNS, sleep, anxiety, QoL) as well as the cognitive functions described in Table 3.

Response:

We have added several paragraphs in the discussion section explain in more detail the association between efavirenz and CNS symptoms such as abnormal dreams.

CNS symptoms have consistently been reported in 40% to 60 % of patients who initiate efavirenz and it is a common reason for discontinuation. However, the majority of CNS side effects appear early and are usually resolved within the first 6 weeks of treatment. There is a proportion of patients were neuropsychiatric symptoms such as sleep difficulties, anxiety and depression persist for much longer periods, even years after initiation of efavirenz therapy (9, 27, 28). The reasons for this phenomenon could be related to interindividual variations in the way the drug is metabolised in the presence or absence of a polymorphism in CYP2BE. Individuals carrying alleles G/T or T/T of CYP2BE are less efficient at metabolising the drug than dose with the allele G/G and therefore are more likely to have a high plasma concentration or longer plasma high life of efavirenz (29). Higher plasma concentration of efavirenz has been associated with CNS symptoms (30)(31). More recently a randomized, double blind study reported that a reduced dose of 400mg of efavirenz was non inferior to the standard dose of 600mg and was associated with a lower frequency of CNS adverse events (32). In our study we were unable to test for individual differences in CYP2BE genotype, and all participants were on the standard 600mg dose of efavirenz which could explain why there were more likely to benefit from a medication switch. There is also the inherent issues with bias associated with patient-reported outcomes measurements in open trials as patients knowledge of the treatment received might influence their view and reporting of their symptoms.”.

In Figure 1, the results are impressive. Could the authors better clarify the causes or correlates of the improved sleep/anxiety/QoL scores? How could the Table 2 results (statistically weak) be logically connected to the Fig 1 results (not weak)? Readers could be interested in individual sample data.

Response:

The results from Table 2 correspond to the questionnaire asking for CNS symptoms based on the DAIDS reporting system. The results presented in Figure 1 are from patient-reported outcome tools that have been validated specifically to assess sleep and anxiety symptoms. They are more sensitive to detect these symptoms that the CNS questionnaire we created to identify all possible CNS symptoms. Therefore, it is likely that the differences in the data are due to the sensitivity of the screening tools. We have expanded on this in the discussion section as a limitation of the study

Because the sample size is relatively small for a clinical trial results, I could recommend showing some data with plots. This could be highly effective for paired data.

Response:

We have added a plot with the results from the depression scores to Figure 1

If statistical analysis was performed with paired tests as described in Methods, I suppose the sample size would be the same regardless of time points. However, the sample size seems to change at different time points (baseline, 4 weeks, 12 weeks, 24 weeks) in Tables 2 and 3. What happens if the unpaired samples are excluded from the statistics described in Tables 2 and 3?

Response:

The reason for changes in the sample size during time points is that some participants drop out of the study. However, all participants who were enrolled and subsequently baselined into the study formed part of the statistical analysis. A sentence explaining this has been added to the statistical section and to the first section in results were we also explained in detail why they drop out of the study.

Figure/Table legends could give more information so that readers will be able to better understand these without referring to the main text.

This has been amended as recommended by reviewer 1 and 2

Discussion seems to be too short and limited. For example, the authors could better discuss the mechanisms for the improved lipid metabolism as described in page 4 lines 167-171, and its impact on the other observations described in Tables 2, 3, and Figure 1.

We have expanded the discussion to address this  and the possible mechanisms for the association of efavirenz and CNS symptoms

I suppose “IQR” mean “interquartile range”, but the abbreviation is not defined in the text. 

Response:

IQR has now been defined

In page 2 line 70, “be” seems unnecessary.

Response:

Line 70 has been removed

In page 3 line 111, some words seem to be missing around “change”.

Response

Amended

Better use of commas could further improve the readability of the text. In Authors’ contributions, I could not identify MS, JJ, and JB. Is MB M. Bracchi or M. Boffito?

Response

Proofreading has been conducted to the whole manuscript address this comment

Round 2

Reviewer 2 Report

The manuscript has been much improved by the revision. The data presentation is not perfect but acceptable. I have just a couple of comments listed below.

Fig 1 data is impressive. However, these are only % changes shown. For the best benefit of readers and science, showing HAD scores with individual plots in addition to % changes could be better. I do not stick to this, and I would like to leave the issue to the authors and the editor. The revised texts might have been written in a hurry. I guess the authors would like to correct mistakes before sending to production.

Author Response

Fig 1 data is impressive. However, these are only % changes shown. For the best benefit of readers and science, showing HAD scores with individual plots in addition to % changes could be better. I do not stick to this, and I would like to leave the issue to the authors and the editor. The revised texts might have been written in a hurry. I guess the authors would like to correct mistakes before sending to production.

Response

We agree with the reviewer comments. We have added a table (Table 3), which contains median IQR and median change values for all patient-reported outcomes presented in Figure 1.

We have proofread the manuscript again to correct any typos of grammatical errors.